# Large-scale optimal transport map estimation using projection pursuit

**Cheng Meng**[1]  **Yuan Ke**[1]  **Jingyi Zhang**[1]  **Mengrui Zhang**[1]  **Wenxuan Zhong**[1]  **Ping Ma**[1]

[1]Department of Statistics, University of Georgia
{cheng.meng25, yuan.ke, jingyi.zhang25, mengrui.zhang, wenxuan, pingma }@uga.edu

## Abstract

This paper studies the estimation of large-scale optimal transport maps (OTM), which is a well known challenging problem owing to the curse of dimensionality. Existing literature approximates the large-scale OTM by a series of one-dimensional OTM problems through iterative random projection. Such methods, however, suffer from slow or none convergence in practice due to the nature of randomly selected projection directions. Instead, we propose an estimation method of large-scale OTM by combining the idea of projection pursuit regression and sufficient dimension reduction. The proposed method, named projection pursuit Monge map (PPMM), adaptively selects the most "informative" projection direction in each iteration. We theoretically show the proposed dimension reduction method can consistently estimate the most "informative" projection direction in each iteration. Furthermore, the PPMM algorithm weakly convergences to the target large-scale OTM in a reasonable number of steps. Empirically, PPMM is computationally easy and converges fast. We assess its finite sample performance through the applications of Wasserstein distance estimation and generative models.

## 1   Introduction

Recently, optimal transport map (OTM) draws great attention in machine learning, statistics, and computer science due to its close relationship to generative models, including generative adversarial nets [19], the "decoder" network in variational autoencoders [27], among others. In a generative model, the goal is usually to generate a "fake" sample, which is indistinguishable from the genuine one. This is equivalent to find a transport map $\phi$ from random noises with distribution $p_X$ (e.g., Gaussian distribution or uniform distribution) to the underlying population distribution $p_Y$ of the genuine sample, e.g., the MNIST or the ImageNet dataset. Nowadays, generative models have been widely-used for generating realistic images [12, 33], songs [4, 13] and videos [32, 53]. Besides generative models, OTM also plays essential roles in various machine learning applications, say color transfer [14, 41], shape match [50], transfer learning [10, 38] and natural language processing [38].

Despite its impressive performance, the computation of OTM is challenging for a large-scale sample with massive sample size and/or high dimensionality. Traditional methods for estimating the OTM includes finding a parametric map and using ordinary differential equations [8, 2]. To address the computational concern, recent developments of OTM estimation have been made based on solving linear programs [44, 37]. Let $\{\boldsymbol{x}_i\}_{i=1}^n \in \mathbb{R}^d$ and $\{\boldsymbol{y}_i\}_{i=1}^n \in \mathbb{R}^d$ be two samples from two continuous probability distributions functions $p_X$ and $p_Y$, respectively. Estimating the OTM from $p_X$ to $p_Y$ by solving a linear program requiring $O(n^3 \log(n))$ computational time for fixed $d$ [38, 47]. To alleviate the computational burden, some literature [11, 17, 1, 21] pursued fast computation approaches of the OTM objective, i.e., the Wasserstein distance. Another school of methods aims to estimate the OTM efficiently when $d$ is small, including multi-scale approaches [35, 18] and dynamic formulations

[48, 36]. These methods utilize the space discretization, thus are generally not applicable in high-dimensional cases.

The random projection method (or known as the radon transformation method) is proposed to estimate OTMs efficiently when $d$ is large [39, 40]. Such a method tackles the problem of estimating a $d$-dimensional OTM iteratively by breaking down the problem into a series of subproblems, each of which finds a one-dimensional OTM using projected samples. Denote $\mathbb{S}^{d-1}$ as the $d$-dimensional unit sphere. In each iteration, a random direction $\boldsymbol{\theta} \in \mathbb{S}^{d-1}$ is picked, and the one-dimensional OTM is then calculated between the projected samples $\{\boldsymbol{x}_i^\mathsf{T}\boldsymbol{\theta}\}_{i=1}^n$ and $\{\boldsymbol{y}_i^\mathsf{T}\boldsymbol{\theta}\}_{i=1}^n$. The collection of all the one-dimensional maps serves as the final estimate of the target OTM. The sliced method modifies the random projection method by considering a large set of random directions from $\mathbb{S}^{d-1}$ in each iteration [7, 42]. The "mean map" of the one-dimensional OTMs over these random directions is considered as a component of the final estimate of the target OTM. We call the random projection method, the sliced method, and their variants as the *projection-based approach*. Such an approach reduces the computational cost of calculating an OTM from $O(n^3 \log(n))$ to $O(Kn \log(n))$, where $K$ is the number of iterations until convergence. However, there is no theoretical guideline on the order of $K$. In addition, the existing projection-based approaches usually require a large number of iterations to convergence or even fail to converge. We speculate that the slow convergence is because a randomly selected projection direction may not be "informative", leading to a one-dimensional OTM that failed to be a decent representation of the target OTM. We illustrate such a phenomenon through an illustrative example as follows.

**An illustrative example.** The left and right panels in Figure 1 illustrates the importance of choosing the "informative" projection direction in OTM estimation. The goal is to obtain the OTM $\phi^*$ which maps a source distribution $p_X$ (colored in red) to a target distribution $p_Y$ (colored in green). For each panel, we first randomly pick a projection direction (black arrow) and

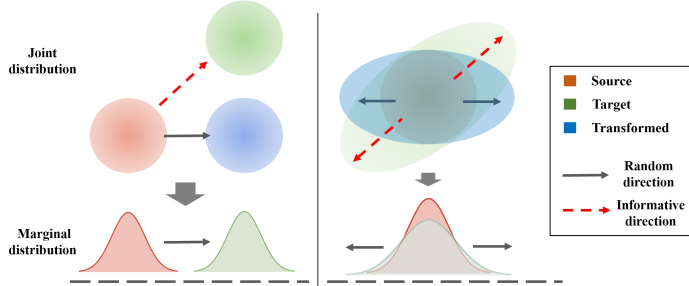

Figure 1: Illustration for the "informative" projection direction

obtain the marginal distributions of $p_X$ and $p_Y$ (the bell-shaped curves), respectively. The one-dimensional OTM then can be calculated based on the marginal distributions. Applying such a map to the source distribution yields the transformed distribution (colored in blue). One can observe that the transformed distributions are significantly different from the target ones. Such an observation indicates that the one-dimensional OTM with respect to a random projection direction may fail to well-represent the target OTM. This observation motivates us to select the "informative" projection direction (red arrow), which yields a better one-dimensional OTM.

**Our contributions.** To address the issues mentioned above, this paper introduces a novel statistical approach to estimate large-scale OTMs. The proposed method, named projection pursuit Monge map (PPMM), improves the existing projection-based approaches from two aspects. First, PPMM uses a sufficient dimension reduction technique to estimate the most "informative" projection direction in each iteration. Second, PPMM is based on projection pursuit [16]. The idea is similar to boosting that search for the next optimal direction based on the residual of previous ones. Theoretically, we show the proposed method can consistently estimate the most "informative" projection direction in each iteration, and the algorithm weakly convergences to the target large-scale OTM in a reasonable number of steps. The finite sample performance of the proposed algorithm is evaluated by two applications: Wasserstein distance estimation and generative model. We show the proposed method outperforms several state-of-the-art large-scale OTM estimation methods through extensive experiments on various synthetic and real-world datasets.

## 2 Problem setup and methodology

**Optimal transport map and Wasserstein distance.** Denote $X \in \mathbb{R}^d$ and $Y \in \mathbb{R}^d$ as two continuous random variables with probability distribution functions $p_X$ and $p_Y$, respectively. The problem of finding a transport map $\phi : \mathbb{R}^d \to \mathbb{R}^d$ such that $\phi(X)$ and $Y$ have the same distribution, has been

widely-studied in mathematics, probability, and economics, see [14, 50, 43] for examples of some new developments. Note that the transport map between the two distributions is not unique. Among all transport maps, it may be of interest to define the "optimal" one according to some criteria. A standard approach, named Monge formulation [52], is to find the OTM[1] $\phi^*$ that satisfies

$$\phi^* = \inf_{\phi \in \Phi} \int_{\mathbb{R}^d} \|X - \phi(X)\|^p \mathrm{d}p_X,$$

where $\Phi$ is the set of all transport maps, $\|\cdot\|$ is the vector norm and $p$ is a positive integer. Given the existence of the Monge map, the Wasserstein distance of order $p$ is defined as

$$W_p(p_X, p_Y) = \left( \int_{\mathbb{R}^d} \|X - \phi^*(X)\|^p \mathrm{d}p_X \right)^{1/p}.$$

Denote $\widehat{\phi}$ as an estimator of $\phi^*$. Suppose one observe $\mathbf{X} = (\boldsymbol{x}_1, \ldots, \boldsymbol{x}_n)^{\mathsf{T}} \in \mathbb{R}^{n \times d}$ and $\mathbf{Y} = (\boldsymbol{y}_1, \ldots, \boldsymbol{y}_n)^{\mathsf{T}} \in \mathbb{R}^{n \times d}$ from $p_X$ and $p_Y$, respectively. The Wasserstein distance $W_p(p_X, p_Y)$ thus can be estimated by

$$\widehat{W}_p(\mathbf{X}, \mathbf{Y}) = \left( \frac{1}{n} \sum_{i=1}^{n} \|\boldsymbol{x}_i - \widehat{\phi}(\boldsymbol{x}_i)\|^p \right)^{1/p}.$$

**Projection pursuit method.** Projection pursuit regression [16, 24, 15, 26] is widely-used for high-dimensional nonparametric regression models which takes the form.

$$z_i = \sum_{j=1}^{s} f_j(\boldsymbol{\beta}_j^{\mathsf{T}} \boldsymbol{x}_i) + \epsilon_i, \quad i = 1, \ldots, n, \tag{1}$$

where $s$ is a hyper-parameter, $\{z_i\}_{i=1}^{n} \in \mathbb{R}$ is the univariate response, $\{\boldsymbol{x}_i\}_{i=1}^{n} \in \mathbb{R}^d$ are covariates, and $\{\epsilon_i\}_{i=1}^{n}$ are i.i.d. normal errors. The goal is to estimate the unknown link functions $\{f_j\}_{j=1}^{s} : \mathbb{R} \to \mathbb{R}$ and the unknown coefficients $\{\boldsymbol{\beta}_j\}_{j=1}^{s} \in \mathbb{R}^d$.

The additive model (1) can be fitted in an iterative fashion. In the $k$th iteration, $k = 2, \ldots, s$, denote $\{(\widehat{f}_j, \widehat{\boldsymbol{\beta}}_j)\}_{j=1}^{k-1}$ the estimate of $\{(f_j, \boldsymbol{\beta}_j)\}_{j=1}^{k-1}$ obtained from previous $k-1$ iterations. Denote $R_i^{[k]} = z_i - \sum_{j=1}^{k-1} \widehat{f}_j(\widehat{\boldsymbol{\beta}}_j^{\mathsf{T}} \boldsymbol{x}_i)$, $i = 1, \ldots, n$, the residuals. Then $(f_k, \boldsymbol{\beta}_k)$ can be estimated by solving the following least squares problem

$$\min_{f_k, \boldsymbol{\beta}_k} \sum_{i=1}^{n} \left[ R_i^{[k]} - f_k(\boldsymbol{\beta}_k^{\mathsf{T}} \boldsymbol{x}_i) \right]^2.$$

The above iterative process explains the intuition behind the projection pursuit regression. Given the model fitted in previous iterations, we fit a one dimensional regression model using the current residuals, rather than the original responses. We then add this new regression model into the fitted function in order to update the residuals. By adding small regression models to the residuals, we gradually improve fitted model in areas where it does not perform well.

The intuition of projection pursuit regression motivates us to modify the existing projection-based OTM estimation approaches from two aspects. First, in the $k$th iteration, we propose to seek a new projection direction for the one-dimensional OTM in the subspace spanned by the residuals of the previously $k-1$ directions. On the contrary, following a direction that is in the span of used ones can lead to an inefficient one dimensional OTM. As a result, this "move" may hardly reduce the Wasserstein distance between $p_X$ and $p_Y$. Such inefficient "moves" can be one of the causes of the convergence issue in existing projection-based OTM estimation algorithms. Second, in each iteration, we propose to select the most "informative" direction with respect to the current residuals rather than a random one. Specifically, we choose the direction that explains the highest proportion of variations in the subspace spanned by the current residuals. Intuitively, this direction addresses the maximum marginal "discrepancy" between $p_X$ and $p_Y$ among the ones that are not considered by previous iterations. We propose to estimate this most "informative" direction with sufficient dimension reduction techniques introduced as follows.

**Sufficient dimension reduction.** Consider a regression problem with univariate response $Z$ and a $d$-dimensional predictor $X$. Sufficient dimension reduction for regression aims to reduce the dimension of $X$ while preserving its regression relation with $Z$. In other words, sufficient dimension reduction seeks a set of linear combinations of $X$, say $\mathbf{B}^\intercal X$ with some $\mathbf{B} \in \mathbb{R}^{d \times q}$ and $q \leq d$, such that $Z$ depends on $X$ only through $\mathbf{B}^\intercal X$, i.e., $Z \perp\!\!\!\perp X | \mathbf{B}^\intercal X$. Then, the column space of $\mathbf{B}$, denoted as $\mathcal{S}(\mathbf{B})$ is called a dimension reduction space (DRS). Furthermore, if the union of all possible DRSs is also a DRS, we call it the central subspace and denote it as $\mathcal{S}_{Z|X}$. When $\mathcal{S}_{Z|X}$ exists, it is the minimum DRS. We call a sufficient dimension reduction method exclusive if it induces a DRS that equals to the central subspace. Some popular sufficient dimension reduction techniques include sliced inverse regression (SIR) [30], principal Hessian directions (PHD) [31], sliced average variance estimator (SAVE) [9], directional regression (DR) [29], among others.

**Estimation of the most "informative" projection direction.** Consider estimating an OTM between a source sample and a target sample. We first form a regression problem by adding a binary response, which equals zero for the source sample and one for the target sample. We then utilize the sufficient dimension reduction technique to select the most "informative" projection direction. To be specific, we select the projection direction $\boldsymbol{\xi} \in \mathbb{R}^d$ as the eigenvector corresponds to the largest eigenvalue of the estimated $\mathbf{B}$. The direction $\boldsymbol{\xi}$ is most "informative" in the sense that, the projected samples $\mathbf{X}\boldsymbol{\xi}$ and $\mathbf{Y}\boldsymbol{\xi}$ have the most substantial " discrepancy." The metric of the "discrepancy" depends on the choice of the sufficient dimension reduction technique. Figure 2 gives a toy example to illustrate this idea. In this paper, we opt to use SAVE for calculating $\mathbf{B}$, and hence

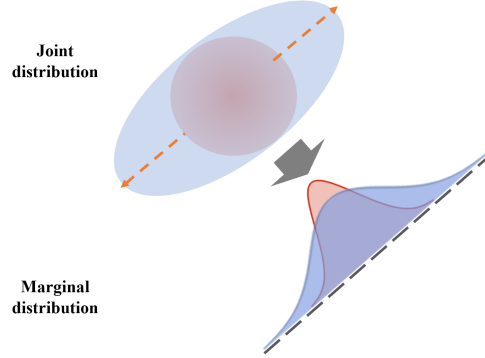

Figure 2: The most "informative" projection direction ensures the projected samples (illustrated by the distributions colored in red and blue, respectively) have the largest "discrepancy".

the "discrepancy" metric is the difference between $\mathrm{Var}(\mathbf{X}\boldsymbol{\xi})$ and $\mathrm{Var}(\mathbf{Y}\boldsymbol{\xi})$. Empirically, we find other sufficient dimension reduction techniques, like PHD and DR, also yield similar performance. The SIR method, however, yields inferior performance, since it only considers the first moment. The Algorithm 1 below introduces our estimation method of "informative" projection direction in detail.

---

**Algorithm 1** Select the most "informative" projection direction using SAVE

**Input:** two standardized matrix $\mathbf{X} \in \mathbb{R}^{n \times d}$ and $\mathbf{Y} \in \mathbb{R}^{n \times d}$

*Step 1:* calculate $\widehat{\boldsymbol{\Sigma}} \in \mathbb{R}^{d \times d}$, i.e., the sample variance-covariance matrix of $\left(\begin{smallmatrix}\mathbf{X}\\\mathbf{Y}\end{smallmatrix}\right)$

*Step 2:* calculate the sample variance-covariance matrices of $\mathbf{X}\widehat{\boldsymbol{\Sigma}}^{-1/2}$ and $\mathbf{Y}\widehat{\boldsymbol{\Sigma}}^{-1/2}$, denoted as $\widehat{\boldsymbol{\Sigma}}_1 \in \mathbb{R}^{d \times d}$ and $\widehat{\boldsymbol{\Sigma}}_2 \in \mathbb{R}^{d \times d}$, respectively

*Step 3:* calculate the eigenvector $\boldsymbol{\xi} \in \mathbb{R}^d$, which corresponding to the largest eigenvalue of the matrix $((\widehat{\boldsymbol{\Sigma}}_1 - I_d)^2 + (\widehat{\boldsymbol{\Sigma}}_2 - I_d)^2)/4$

**Output:** the final result is given by $\widehat{\boldsymbol{\Sigma}}^{-1/2}\boldsymbol{\xi}/||\widehat{\boldsymbol{\Sigma}}^{-1/2}\boldsymbol{\xi}||$, where $||\cdot||$ denotes the Euclidean norm

---

**Projection pursuit Monge map algorithm.** Now, we are ready to present our estimation method for large-scale OTM. The detailed algorithm, named projection pursuit Monge map, is summarized in Algorithm 2 below. In each iteration, the PPMM applies a one-dimensional OTM following the most "informative" projection direction selected by the Algorithm 1.

**Computational cost of PPMM.** In Algorithm 2, the computational cost mainly resides in the first two steps within each iteration. In step (a), one calculates $\boldsymbol{\xi}_k$ using Algorithm 1, whose computational cost is of order $O(nd^2)$. In step (b), one calculates a one-dimensional OTM using the look-up table, which is simply a sorting algorithm [40, 38].

The computational cost for step (b) is of order $O(n\log(n))$. Suppose that the algorithm converges after $K$ iterations. The overall computational cost of Algorithm 2 is of order $O\left(Knd^2 + Kn\log(n)\right)$. Empirically, we find $K = O(d)$ works reasonably well. When $\log(n)^{1/2} \leq d \ll n^{2/3}$, the order of computational cost of PPMM is $o\left(n^3 \log(n)\right)$ which is smaller than the computational cost of

---
**Algorithm 2** Projection pursuit Monge map (PPMM)

---
**Input:** two matrix $\mathbf{X} \in \mathbb{R}^{n \times d}$ and $\mathbf{Y} \in \mathbb{R}^{n \times d}$
$k \leftarrow 0, \mathbf{X}^{[0]} \leftarrow \mathbf{X}$
**repeat**
    (a) calculate the projection direction $\boldsymbol{\xi}_k \in \mathbb{R}^d$ between $\mathbf{X}^{[k]}$ and $\mathbf{Y}$ (using Algorithm 1)
    (b) find the one-dimensional OTM $\phi^{(k)}$ that matches $\mathbf{X}^{[k]}\boldsymbol{\xi}_k$ to $\mathbf{Y}\boldsymbol{\xi}_k$ (using look-up table)
    (c) $\mathbf{X}^{[k+1]} \leftarrow \mathbf{X}^{[k]} + (\phi^{(k)}(\mathbf{X}^{[k]}\boldsymbol{\xi}_k) - \mathbf{X}^{[k]}\boldsymbol{\xi}_k)\boldsymbol{\xi}_k^{\mathsf{T}}$ and $k \leftarrow k+1$
**until** converge
The final estimator is given by $\widehat{\phi} : \mathbf{X} \to \mathbf{X}^{[k]}$

---

the naive method for calculating OTMs. When $d \leq \log(n)^{1/2}$, the order of computational cost reduces to $O\left(Kn \log(n)\right)$ which is faster than the exiting projection-based methods given PPMM converges faster. The memory cost for Algorithm 2 mainly resides in the step (a), which is of the order $O(Knd^2)$.

## 3 Theoretical results

**Exclusiveness of SAVE.** For mathematical simplicity, we assume $E[X] = E[Y] = \mathbf{0}_d$. When $E[X] \neq E[Y]$, one can use a first-order dimension reduction method like SIR to adjust means before applying SAVE.

Denote $W = (X+Y)/2$, $\Sigma_W = \text{Var}(W)$, and $Z = W\Sigma_W^{-1/2}$. For a univariate continuous response variable $R$, one can approximate the central subspace $\mathcal{S}_{R|Z}$ by $\mathcal{S}_{\text{SAVE}}$, which is the population version of the dimension reduction space of SAVE. To be specific, $\mathcal{S}_{\text{SAVE}}$ is the column space of matrix

$$E[\text{Var}(Z|R) - I_d]^2 = \frac{1}{4}\left\{E[\text{Var}(X\Sigma_W^{-1/2}|R) - I_d]^2 + E[\text{Var}(Y\Sigma_W^{-1/2}|R) - I_d]^2\right\},$$

where the above equation used the fact that $X \perp\!\!\!\perp Y$.

**Assumption 1.** *Let $P$ be the projection onto the central space $\mathcal{S}_{R|Z}$ with respect to the inner project $a \cdot b = a^{\mathsf{T}}b$. For any nonzero vectors $u, v \in \mathbb{R}^d$, such that $u$ is orthogonal to $\mathcal{S}_{R|Z}$ and $v \in \mathcal{S}_{R|Z}$, we assume*

    *(a) $E(u^{\mathsf{T}}Z|PZ)$ is a linear function of Z;*

    *(b) $\text{Var}(u^{\mathsf{T}}Z|PZ)$ is a nonrandom number;*

    *(c) Let $(\widetilde{Z}, \widetilde{R})$ be an independent copy of $(Z, R)$. $E\left[v^{\mathsf{T}}(Z - \widetilde{Z})^2|R, \widetilde{R}\right]$ is non degenerate; that is, it is not equal almost surely to a constant.*

**Theorem 1.** *Let $R$ be a univariate continuous response variable. Under Assumption 1, the dimension reduction space induced by SAVE is exclusive. In other words, $\mathcal{S}_{\text{SAVE}} = \mathcal{S}_{R|Z}$.*

**Consistency of the most "informative" projection direction.** Let $\widehat{\Sigma}_1$ and $\widehat{\Sigma}_2$ be the sample covariance matrix estimator of $\Sigma_1$ and $\Sigma_2$, respectively. Denote

$$\Sigma_{\text{SAVE}} = \frac{1}{4}\left[(\Sigma_1 - I_d)^2 + (\Sigma_2 - I_d)^2\right] \quad \text{and} \quad \widehat{\Sigma}_{\text{SAVE}} = \frac{1}{4}\left[(\widehat{\Sigma}_1 - I_d)^2 + (\widehat{\Sigma}_2 - I_d)^2\right].$$

Denote $\boldsymbol{\xi}_1$ and $\widehat{\boldsymbol{\xi}}_1$ the eigenvectors correspond to the largest eigenvalues of $\Sigma_{\text{SAVE}}$ and $\widehat{\Sigma}_{\text{SAVE}}$, respectively. Further, denote $r = \text{Rank}(\Sigma_{\text{SAVE}})$, the rank of $\Sigma_{\text{SAVE}}$.

**Assumption 2.** *Let $\{\boldsymbol{x}_i, \boldsymbol{y}_i\}_{i=1}^n$ be an i.i.d. sample of $(X, Y)$. We assume that*

    *(a) Denote $x_{ij}$ and $y_{ik}$ the jth and kth component of $\boldsymbol{x}_i$ and $\boldsymbol{y}_i$, respectively. $E(x_{ij}y_{ik}) = 0$ for all $1 \leq i \leq n$ and $1 \leq j, k \leq d$;*

    *(b) There are $r_1, r_2 > 0$ and $b_1, b_2 > 0$ such that, for any $s > 0$, $1 \leq i \leq n$ and $1 \leq j \leq d$,*

$$P(|x_{ij}| > s) \leq \exp\left\{-(s/b_1)^{r_1}\right\} \quad \text{and} \quad P(|y_{ij}| > s) \leq \exp\left\{-(s/b_2)^{r_2}\right\};$$

*(c) Let $\lambda_1 \ldots, \lambda_d$ be the eigenvalues of $\Sigma_{\text{SAVE}}$ in descending order. There exist positive constants $c_l$ $c_u$ and $c_3$ such that*

$$c_l \leq \min_{1 \leq l \leq r-1}(\lambda_l - \lambda_{l+1})d^{-1/2} \leq c_u, \quad and \quad 0 \leq \lambda_{r+1} < c_3.$$

Theorem 2 shows that Algorithm 1 can consistently estimate the most "informative" projection direction. The $O_p$ in Theorem 2 stands for order in probability, which is similar to $O$ but for random variables.

**Theorem 2.** *Under Assumption 2, the SAVE estimator of most "informative" projection direction satisfies,*

$$\|\widehat{\boldsymbol{\xi}}_1 - \boldsymbol{\xi}_1\|_\infty = O_p(r^4\sqrt{\frac{\log d}{n}} + r^4\sqrt{d}\frac{\log d}{n}), \quad as \quad n, d \to \infty.$$

**Weak convergence of PPMM algorithm.** Denote $\phi^*$ as the $d$-dimensional optimal transport map from $p_X$ to $p_Y$ and $\phi^{(K)}$ as the PPMM estimator after $K$ iterations, i.e. $\phi^{(K)}(\mathbf{X}) = \mathbf{X}^{[K]}$. The following theorem gives the weak convergence results of the PPMM algorithm.

**Theorem 3.** *Suppose Assumption 1 and Assumption 2 hold. Let $K \geq Cd$ for some large enough positive constant $C$, one has*

$$\widehat{W}_p\Big(\phi^{(K)}(\mathbf{X}), \mathbf{X}\Big) \to W_p\Big(\phi^*(X), X\Big), \quad and \quad \phi^{(K)}(\mathbf{X}) \to \phi^*(X) \quad as \quad n \to \infty.$$

Works are proving the convergence rates of the empirical optimal transport objectives [5, 49, 6, 54]. The convergence rate of the OTM has rarely been studied except for a recent paper [25]. We believe Theorem 3 is the first step in this direction.

## 4 Numerical experiments

### 4.1 Estimation of optimal transport map

Suppose that we observe i.i.d. samples $\mathbf{X} = (\boldsymbol{x}_1, \ldots, \boldsymbol{x}_n)^\intercal$ from $p_X = \mathcal{N}_d(\boldsymbol{\mu}_X, \boldsymbol{\Sigma}_X)$ and $\mathbf{Y} = (\boldsymbol{y}_1, \ldots, \boldsymbol{y}_n)^\intercal$ from $p_Y = \mathcal{N}_d(\boldsymbol{\mu}_Y, \boldsymbol{\Sigma}_Y)$, respectively. We set $n = 10,000$, $d = \{10, 20, 50\}$, $\boldsymbol{\mu}_X = -2, \boldsymbol{\mu}_Y = 2, \boldsymbol{\Sigma}_X = 0.8^{|i-j|}$, and $\boldsymbol{\Sigma}_Y = 0.5^{|i-j|}$, for $i, j = 1, \ldots, d$.

We apply PPMM to estimate the OTM between $p_X$ and $p_Y$ from $\{\boldsymbol{x}_i\}_{i=1}^n$ and $\{\boldsymbol{y}_i\}_{i=1}^n$. In comparison, we also consider the following two projection-based competitors: (1) the random projection method (RANDOM) as proposed in [39, 40]; (2) the sliced method as proposed in [7, 42]. The number of slices $L$ is set to be 10, 20, and 50. We assess the convergence of each method by the estimated Wasserstein distance of order 2 after each iteration, i.e. $\widehat{W}_2\Big(\phi^{(k)}(\mathbf{X}), \mathbf{X}\Big)$, where $\phi^{(k)}(\cdot)$ is the estimator of OTM after $k$th iteration. For all three methods, we set the maximum number of iterations to be 200. Notice that, the Wasserstein distance between $p_X$ and $p_Y$ admits a closed form,

$$W_2^2(p_X, p_Y) = \|\boldsymbol{\mu}_X - \boldsymbol{\mu}_Y\|_2^2 + trace\left(\boldsymbol{\Sigma}_X + \boldsymbol{\Sigma}_Y - 2(\boldsymbol{\Sigma}_X^{1/2}\boldsymbol{\Sigma}_Y\boldsymbol{\Sigma}_X^{1/2})^{1/2}\right), \tag{2}$$

which serves as the ground-truth. The results are presented in Figure 3.

In all three scenarios, PPMM (red line) converges to the ground truth within a small number of iterations. The fluctuations of the convergence curves observed in Figure 3 are caused by the non-equal sample means. This can be adjusted by applying a first-order dimension reduction method (e.g., SIR). We do not pursue this approach as the fluctuations do not cover the main pattern in Figure 3. When $d = 10$, RANDOM and SLICED converge to the ground truth but in a much

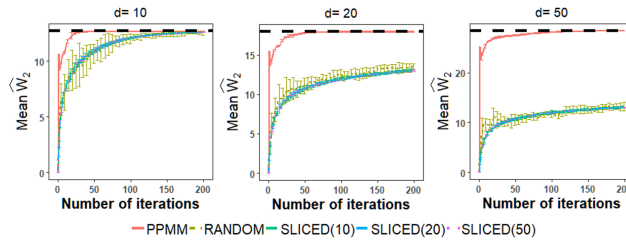

Figure 3: The black dashed line is the true value of the Wasserstein distance as in (2). The colored lines represent the sample mean of the estimated Wasserstein distances over 100 replications, and the vertical bars represent the standard deviations.

slower manner. When $d = 20$ and 50, neither RANDOM nor SLICED manages to converge within 200 iterations. We also find a large number of slices $L$ does not necessarily lead to a better estimation for the SLICED method. As we can see, PPMM is the only one among three that is adaptive to large-scale OTM estimation problems.

In Table 1 below, we compare the computational cost of three methods by reporting the CPU time per iteration over 100 replication.[2] As we expected, the RANDOM method has the lowest CPU time per iteration due to it does not select projection direction. We notice that the CPU time per iteration of the SLICED method is proportional to the number of slices $L$. Last but not least, the CPU time per iteration of PPMM is slightly larger than RANDOM but much smaller than SLICED.

Table 1: The mean CPU time (sec) per iteration, with standard deviations presented in parentheses

|  | PPMM | RANDOM | SLICED(10) | SLICED(20) | SLICED(50) |
|---|---|---|---|---|---|
| $d = 10$ | 0.019 (0.008) | 0.011 (0.008) | 0.111 (0.019) | 0.213 (0.024) | 0.529 (0.031) |
| $d = 20$ | 0.027 (0.011) | 0.014 (0.008) | 0.125 (0.027) | 0.247 (0.033) | 0.605 (0.058) |
| $d = 50$ | 0.059 (0.036) | 0.015 (0.008) | 0.171 (0.037) | 0.338 (0.049) | 0.863 (0.117) |

In the Table 2 below, we report the mean convergence time over 100 replications for PPMM, RANDOM, SLICED, the refined auction algorithm (AUCTIONBF)[3], the revised simplex algorithm (REVSIM) [34] and the shortlist method (SHORTSIM) [20].[3] Table 2 shows that the PPMM is the most computationally efficient method thanks to its cheap per iteration cost and fast convergence.

Table 2: The mean convergence time (sec) for estimating the Wasserstein distance, with standard deviations presented in parentheses. The symbol "-" is inserted when the algorithm fails to converge.

|  | PPMM | RANDOM | SLICED(10) | AUCTIONBF | REVSIM | SHORTSIM |
|---|---|---|---|---|---|---|
| $d = 10$ | 0.6 (0.1) | 4.8 (1.7) | 23.0 (2.6) | 99.7 (10.4) | 40.2 (4.0) | 42.5 (3.2) |
| $d = 20$ | 2.1 (0.3) | 24.4 (3.2) | 230.2 (28.4) | 109.4 (12.5) | 42.6 (5.3) | 50.2 (6.6) |
| $d = 50$ | 5.5 (0.4) | - | - | 125.5 (13.3) | 46.5 (5.6) | 56.5 (7.1) |

## 4.2 Application to generative models

A critical issue in generative models is the so-called mode collapse, i.e., the generated "fake" sample fails to capture some modes present in the training data [22, 45]. To address this issue, recent studies [51, 22, 28] incorporated generative models with the optimal transportation theory. As illustrated in Figure 4, one can decompose the problem of generating fake samples into two major steps: (1) manifold learning and (2) probability transformation. The step (1) aims to discover the manifold structure of the training data by mapping the training data from the original space $\mathcal{X} \subset \mathbb{R}^d$ to a latent space $\mathcal{Z} \subset \mathbb{R}^{d^*}$ with $d^* \ll d$. Notice that the probability distribution of the transformed data in $\mathcal{Z}$ may not be convex, leading to the problem of mode collapse. The step (2) then addresses the mode collapse issue through transporting the distribution in $\mathcal{Z}$ to the uniform distribution $U([0, 1]^{d^*})$. Then, the generative model takes a random input from $U([0, 1]^{d^*})$ and sequentially applies the inverse transformations in step (2) and step (1) to generate the output. In practice, one may implement the step (1) and (2) using variational autoencoders (VAE) and OTM, respectively. As we can see, the estimation of OTM plays an essential role in this framework.

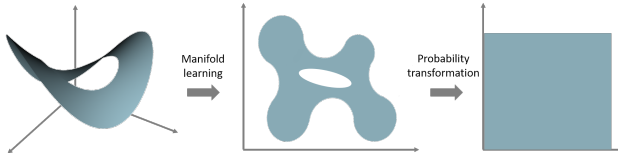

Figure 4: Illustration for the generative model using manifold learning and optimal transport

In this subsection, we apply PPMM as well as RANDOM and SLICED to generative models to study two datasets: MINST and Google doodle dataset. For the SLICED method, we set the number of slices to be 10, 20, and 50. For all three methods, we set the number of iterations is set to be $10d^*$. We use the squared Euclidean distance as the cost for the VAE model.

Table 3: The FID for the generated samples (lower the better), with standard deviations presented in parentheses

|  | PPMM | RANDOM | SLICED(10) | SLICED(20) | SLICED(50) |
|---|---|---|---|---|---|
| MNIST | **0.17** (0.01) | 4.62 (0.02) | 2.98 (0.01) | 3.04 (0.01) | 3.12 (0.01) |
| Doodle (face) | **0.59** (0.09) | 8.78 (0.04) | 5.69 (0.01) | 6.01 (0.01) | 5.52 (0.01) |
| Doodle (cat) | **0.24** (0.03) | 8.93 (0.03) | 5.99 (0.01) | 5.26 (0.01) | 5.33 (0.01) |
| Doodle (bird) | **0.36** (0.03) | 7.81 (0.03) | 5.44 (0.01) | 5.50 (0.01) | 4.98 (0.01) |

**MNIST.** We first study the MNIST dataset, which contains 60,000 training images and 10,000 testing images of hand written digits. We pull each $28 \times 28$ image to a 784-dimensional vector and rescale the grayscale values from $[0, 255]$ to $[0, 1]$. Following the method in [51], we apply VAE to encode the data into a latent space $\mathcal{Z}$ of dimensionality $d^* = 8$. Then, the OTM from the distribution in $\mathcal{Z}$ to $U([0, 1]^8)$ is estimated by PPMM as well as RANDOM and SLICED.

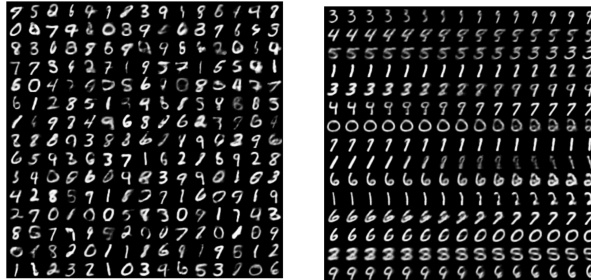

Figure 5: Left: random samples generated by PPMM. Right: linear interpolation between random pairs of images.

First, we visually examine the fake sample generated with PPMM. In the left-hand panel of Figure 5, we display some random images generated by PPMM. The right-hand panel of Figure 5 shows that PPMM can predict the continuous shift from one digit to another. To be specific, let $\boldsymbol{a}, \boldsymbol{b} \in \mathbb{R}^{784}$ be the sample of two digits (e.g. 3 and 9) in the testing set. Let $T : \mathcal{X} \to \mathcal{Z}$ be the map induced by VAE and $\widehat{\phi}$ the OTM estimated by PPMM. Then, $\widehat{\phi}(T(\cdot))$ maps the sample distribution to $U([0, 1]^8)$. We linearly interpolate between $\widehat{\phi}(T(\boldsymbol{a}))$ and $\widehat{\phi}(T(\boldsymbol{b}))$ with equal-size steps. Then we transform the interpolated points back to the sample distribution to generate the middle columns in the right panel of Figure 5.

We use the *"Fréchet Inception Distance"* (FID) [23] to quantify the similarity between the generated fake sample and the training sample. Specifically, we first generate 1,000 random inputs from $U([0, 1]^8)$. We then apply PPMM, RANDOM, and SLICED to this input sample, yields the fake samples in the latent space $\mathcal{Z}$. Finally, we calculate the FID between the encoded training sample in the latent space and the generated fake samples, respectively. A small value of FID indicates the generated fake sample is similar to the training sample and vice versa. The sample mean and sample standard deviation (in parentheses) of FID over 50 replications are presented in Table 3. Table 3 indicates PPMM significantly outperforms the other two methods in terms of estimating the OTM.

**Google doodle dataset.** The Google Doodle dataset[4] contains over 50 million drawings created by users with a mouse under 20 secs. We analyze a pre-processed version of this dataset from the quick draw Github account[5]. In the dataset we use, the drawings are centered and rendered into $28 \times 28$ grayscale images. We pull each $28 \times 28$ image to a 784-dimensional vector and rescale the grayscale values from $[0, 255]$ to $[0, 1]$. In this experiment, we study the drawings from three different categories: smile face, cat, and bird. These three categories contain 161,666, 123,202, and 133,572 drawings, respectively. Within each category, we randomly split the data into a training set and a validation set of equal sample sizes.

We apply VAE to the training set with a stopping criterion selected by the validation set. The dimension of the latent space is set to be 16. Let $\boldsymbol{a}, \boldsymbol{b} \in \mathbb{R}^{784}$ be two vectors in the validation set, $T : \mathcal{X} \to \mathcal{Z}$ be the map induced by VAE and $\widehat{\phi}$ be the OTM estimated by PPMM. Note that $\widehat{\phi}(T(\cdot))$ maps the sample distribution to $U([0, 1]^{16})$. We then linearly interpolate between $\widehat{\phi}(T(\boldsymbol{a}))$ and $\widehat{\phi}(T(\boldsymbol{b}))$ with equal-size steps. The results are presented in Figure 6.

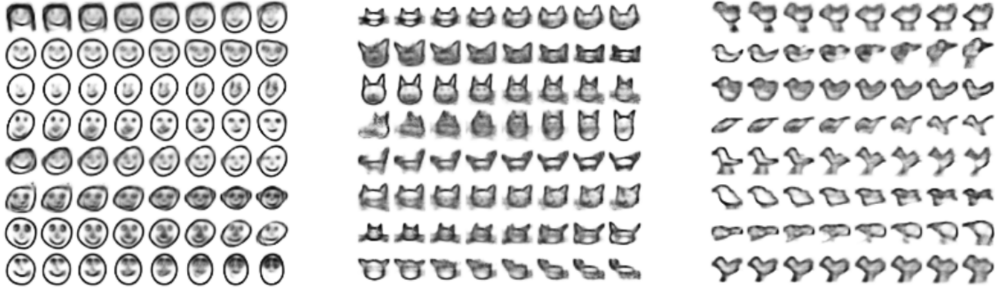

Figure 6: Linear interpolation between random pairs of images from the dataset of smile face (left), cat (center), and bird (right).

Then, we quantify the similarity between the generated fake samples and the truth by calculating the FID in the latent space. The sample mean and sample standard deviation (in parentheses) of FID over 50 replications are presented in Table 3. Again, the results in Table 3 justify the superior performance of PPMM over existing projection-based methods.

# 5 Extensions

First, the PPMM can be extended to address the penitential heterogeneous in the dataset by assigning non-equal weights to the points in source and target samples. This is equivalent to calculate weighted variance-covariance matrices in Step 2 of Algorithm 1. Second, the PPMM method can be modified to allow the sizes of the source and target samples to be different. In such a scenario, we can replace the look-up table in the Step (b) of Algorithm 2 with an approximate lookup table. Recall that the one-dimensional lookup table is just sorting, the one-dimensional approximate look-up table can be achieved by combining sorting and linear interpolation. We validate the above extensions with a simulated experiment similar to the one in Section 4.1 except that we draw $5,000$ and

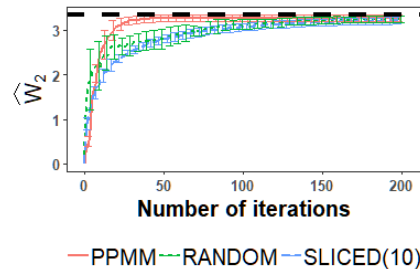

Figure 7: Experiment for heterogeneous data with non-equal sample sizes. The black dashed line is the oracle calculated by SHORTSIM

$1,000$ points from $p_X$ and $p_Y$, respectively. We set $d = 10$ and assign weights to the observations randomly. The estimation results are presented in Figure 7. In addition, the average convergence time is: PPMM(0.3s), RANDOM (1.4s), SLICED10 (14s) and SHORTSIM (74s).

Theorem 3 suggests that, for the PPMM algorithm, the number of iterations until converge, i.e., $K$, is on the order of dimensionality $d$. Here we use a simulated example to assess whether this order is attainable. We follow a similar setting as in Section 4.1 except that we increase $d$ from 10 to 100 with a step size of 10. Besides, we set the termination criteria to be a hard threshold, i.e., $10^{-5}$. In Figure 8, we report the sample mean (solid line) and standard deviation (vertical bars) of $K$ over 100 replications with respect to the increased $d$. One can observe a clear linear pattern.

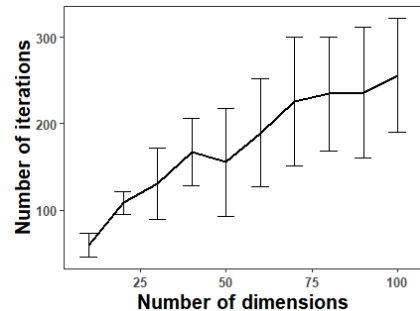

Figure 8: Number of iterations to converge

**Acknowledgment**

We would like to thank Xiaoxiao Sun, Rui Xie, Xinlian Zhang, Yiwen Liu, and Xing Xin for many fruitful discussions. We would also like to thank Dr. Xianfeng David Gu for his insightful blog about the Optimal transportation theory. Also, we would like to thank the UC Irvine Machine Learning Repository for dataset assistance. This work was partially supported by National Science Foundation grants DMS-1440037, DMS-1440038, DMS-1438957, and NIH grants R01GM113242, R01GM122080.

## Footnotes

[1]Such a map is thus also called the Monge map.

[2]The experiments are implemented by an Intel 2.6 GHz processor.

[3]AUCTIONBF, REVSIM and SHORTSIM are implemented by the R package "transport" [46].

[4]https://quickdraw.withgoogle.com/data

[5]https://github.com/googlecreativelab/quickdraw-dataset

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
