[Supplementary Material · (NeurIPS2019)Appendix_Large_scale_optimal_transport_map_approximation_using_projection_pursuit.pdf]

# Appendix to "Large-scale optimal transport map estimation using projection pursuit"

## 1 A Appendix

2 This appendix provides the proofs of the theoretical results for the main document.

### 3 A.1 Proof of Theorem 1

4 First, we presents some Lemmas to facilitate the proof of Theorem 1.

5 Let $(\widetilde{Z}, \widetilde{R})$ be an independent copy of $(Z, R)$. We denote

$$A(R, \widetilde{R}) = E\left[(Z - \widetilde{Z})(Z - \widetilde{Z})^{\intercal} | R, \widetilde{R}\right]. \tag{1}$$

Let $P$ be the projection onto the central space $\mathcal{S}_{R|Z}$ with respect to the inner project $a \cdot b = a^{\intercal} b$, and let $Q = I_d - P$. Further, define two quantities

$$C = 2I_d - A(R, \widetilde{R}) \quad \text{and} \quad G = E(C)^2.$$

6 **Lemma 1.** *Denote* $\mathrm{span}(G)$ *the column space of matrix G, then* $\mathcal{S}_{SAVE} = span(G)$.

7 *Proof of Lemma 1.* Follow the Theorem 2 in [4] and notice $E(ZZ^{\intercal}) = I_d$, the matrix $G$ can be
8 re-expressed as

$$\begin{aligned} G =& 2E\left[E^2(ZZ^{\intercal} - I_d | R)\right] + 2E^2\left[E(Z|R)E(Z^{\intercal}|R)\right] \\ &+ 2E\left[E(Z^{\intercal}|R)E(Z|R)\right] E\left[E(Z|R)E(Z^{\intercal}|R)\right]. \end{aligned}$$

9 First, let $v$ be a vector orthogonal to $\mathcal{S}_{\text{SAVE}}$. We have $E(Z^{\intercal}|R)v = 0$ and $[I_d - \mathrm{var}(Z|R)]v = 0$
10 almost surely. Therefore, $G_i v = 0$ for $i = 1, \dots, 6$. This implies that $v$ is orthogonal to $\mathrm{span}(G)$,
11 and hence $\mathrm{span}(G) \subseteq \mathcal{S}_{\text{SAVE}}$.

12 On the other hand, let $v$ be a vector orthogonal to $\mathrm{span}(G)$. Then, $v^{\intercal} G v = 0$ implies

$$v^{\intercal} E\left[E^2(ZZ^{\intercal} - I_d | R)\right] v = 0 \tag{2}$$

13 and

$$v^{\intercal} E\left[E(Z^{\intercal}|R)E(Z|R)\right] E\left[E(Z|R)E(Z^{\intercal}|R)\right] v = 0, \tag{3}$$

14 almost surely.

15 The second equality implies that $E(Z^{\intercal}|R) = 0$ almost surely. Furthermore, Using the fact that
16 $E(ZZ^{\intercal}) = I_d$ and $E(ZZ^{\intercal}|R) = \mathrm{var}(Z|R) + E(Z|R)E(Z^{\intercal}|Y)$, the first inequality can be re-
17 expressed as

$$\begin{aligned} 0 =& v^{\intercal} E\left[\mathrm{var}(Z|R) - I_d\right]^2 v \\ &+ v^{\intercal} E\left[(\mathrm{var}(Z|R) - I_p)E(Z|R)E(Z^{\intercal}|R)\right] v \\ &+ v^{\intercal} E\left[E(Z|R)E(Z^{\intercal}|R)(\mathrm{var}(Z|R) - I_d)\right] v \\ &+ v^{\intercal} E\left[E(Z|R)E(Z^{\intercal}|R)\right]^2 v. \end{aligned}$$

The second to fourth terms are 0 since $E(Z^\mathsf{T}|R) = 0$. Thus the first term must also be 0, almost surely, implying . that $v \perp\!\!\!\perp \mathcal{S}_{\text{SAVE}}$. We complete the proof by showing that $\mathcal{S}_{\text{SAVE}} \subseteq \text{span}(G)$.

$\square$

**Lemma 2.** *Suppose the Assumption 1 (a) and (b) hold. Denote* $\text{span}(G)$ *the column space of matrix* $G$, *then* $\mathcal{S}_{SAVE} = span(G)$.

*Proof of Lemma 2.* By Lemma 2.1 of [5] and Propsition 4.6 of [1], $(Z, R) \perp\!\!\!\perp (\widetilde{Z}, \widetilde{R})$ implies that $Z \perp\!\!\!\perp \widetilde{Z}(R, \widetilde{R})$, $Z \perp\!\!\!\perp \widetilde{R}|R$ and $\widetilde{Z} \perp\!\!\!\perp R|\widetilde{R}$. Thus $A(R, \widetilde{R})$ can be re-expressed as

$$\begin{aligned} A(R, \widetilde{R}) =& E(ZZ^\mathsf{T}|R) - E(Z|R)E(\widetilde{Z}^\mathsf{T}|\widetilde{R}) \\ & -E(\widetilde{Z}|\widetilde{R})E(Z^\mathsf{T}|R) + E(\widetilde{Z}\widetilde{Z}^\mathsf{T}|\widetilde{R})) \end{aligned} \tag{4}$$

Let $v$ be a vector orthogonal to $\mathcal{S}_{R|W}$. By assumption (a), $E(v^\mathsf{T}Z|PZ) = \alpha^\mathsf{T}PZ$ for some $\alpha \in \mathbb{R}^d$. Multiply both sides by $ZP\alpha$ and then take unconditional expectation to obtain $v^\mathsf{T}P\alpha = \alpha^\mathsf{T}P\alpha = 0$. Thus $E(v^\mathsf{T}Z|PZ) = 0$.

By Assumption 1 (a) and (b), $E\left[(v^\mathsf{T}Z)^2|PZ\right] = c + E^2(v^\mathsf{T}Z|PZ) = c$, for some constant $c$. Take unconditional expectations on both sides to obtain $c = v^\mathsf{T}v$. Thus $E\left[(v^\mathsf{T}Z)^2|PZ\right] = v^\mathsf{T}v$.

Because $R \perp\!\!\!\perp Z|PZ$, we have

$$\begin{aligned} E(v^\mathsf{T}Z|R) &= E\left[E(v^\mathsf{T}Z|PZ|R)\right] = 0, \\ E\left[(v^\mathsf{T}Z)^2|R\right] &= E\left\{E[(v^\mathsf{T}Z)^2|PZ]|R\right\} = v^\mathsf{T}v. \end{aligned}$$

Substitute the above two lines into 4, we have

$$v^\mathsf{T}A(R, \widetilde{R})v = 2v^\mathsf{T}v,$$

which implies $v^\mathsf{T}Gv = 0$. Then, we have $\text{span}(G) \subseteq \mathcal{S}_{R|W}$.

$\square$

**Lemma 3.** *Let $G$ be a symmetric and positive semi-definite matrix which satisfies* $\text{span}(G) \subseteq \mathcal{S}_{R|W}$. *Then,* $\text{span}(G) = \mathcal{S}_{R|W}$ *iff* $v^\mathsf{T}Gv > 0$ *for all* $v \in \mathcal{S}_{R|W}$, $v \neq 0$.

*Proof of Lemma 3.* Suppose that $\text{span}(G)$ is a strict subspace of $\mathcal{S}_{R|W}$. Then $v^\mathsf{T}Gv = 0$ for any $v \neq 0$, $v \in \mathcal{S}_{R|W} \ominus \text{span}(G)$. Conversely, for $\text{span}(G) = \mathcal{S}_{R|W}$, $v \in \mathcal{S}_{R|W}$, $v \neq 0$, we have $v \in \text{span}(G)$, and hence $v^\mathsf{T}Gv > 0$. $\square$

**Proof of Theorem 1.** We first show that $\text{span}(G) = \mathcal{S}_{R|W}$. $G$ is symmetric and positive semi-definite according to its definition. Also, Lemma 2 shows $\text{span}(G) \subseteq \mathcal{S}_{R|W}$ under Assumption 1 (a) and (b).

Let $v \in \mathcal{S}_{R|W}$, $v \neq 0$. Without loss of generality, we assume $\|v\| = 1$. Then

$$v^\mathsf{T}Gv = v^\mathsf{T}E\left[C(I_d - vv^\mathsf{T})C\right]v + E\left[(v^\mathsf{T}Cv)^2\right]. \tag{5}$$

Because $I_d - vv^\mathsf{T} \geq 0$, the first term on the right hand side of (5) is nonnegative. By Assumption 1 (c), $v^\mathsf{T}A(R, \widetilde{R})v$ is non-degenerate. Therefore, $v^\mathsf{T}Cv$ is non-degenerate. Then, by Jensen's inequality and notice $E(C) = 0$,

$$E\left[(v^\mathsf{T}Cv)^2\right] > \left[E(v^\mathsf{T}Cv)\right]^2 = 0. \tag{6}$$

Then, by Lemma 1 and Lemma 3, we complete the proof by showing $\mathcal{S}_{\text{SAVE}} = \text{span}(G) = \mathcal{S}_{R|W}$.

$\square$

## A.2 Proof of Theorem 2

**Proof of Theorem 2.** Suppose Assumption 2 holds. By applying Theorem 3 and Proposition 3 in [2], we arrive at

$$
\begin{aligned}
\|\widehat{\boldsymbol{\xi}}_1 - \boldsymbol{\xi}_1\|_\infty &\leq \max_{1 \leq l \leq r} \|\widehat{\boldsymbol{\xi}}_l - \boldsymbol{\xi}_l\|_\infty \\
&\leq C_1 d^{-3/2} (r^4 \|\widehat{\Sigma}_{\mathrm{SAVE}} - \Sigma_{\mathrm{SAVE}}\|_\infty + r^{3/2} \|\widehat{\Sigma}_{\mathrm{SAVE}} - \Sigma_{\mathrm{SAVE}}\|_2) \\
&\leq C_2 r^4 d^{-1/2} \|\widehat{\Sigma}_{\mathrm{SAVE}} - \Sigma_{\mathrm{SAVE}}\|_{\max},
\end{aligned}
\tag{7}
$$

where $C_1$ and $C_2$ are some positive constants.

It can be shown that

$$
\begin{aligned}
\widehat{\Sigma}_{\mathrm{SAVE}} &- \Sigma_{\mathrm{SAVE}} \\
&= \frac{1}{4} \left[ (\widehat{\Sigma}_1 - I_d)^2 - (\Sigma_1 - I_d)^2 + (\widehat{\Sigma}_2 - I_d)^2 - (\Sigma_2 - I_d)^2 \right] \\
&= \frac{1}{4} \left[ (\widehat{\Sigma}_1 + \Sigma_1 - 2I_d)(\widehat{\Sigma}_1 - \Sigma_1) + (\widehat{\Sigma}_2 + \Sigma_2 - 2I_d)(\widehat{\Sigma}_2 - \Sigma_2) \right]
\end{aligned}
$$

Then,

$$
\begin{aligned}
\|\widehat{\Sigma}_{\mathrm{SAVE}} &- \Sigma_{\mathrm{SAVE}}\|_{\max} \\
&\leq \frac{1}{4} \left[ \|(\widehat{\Sigma}_1 + \Sigma_1 - 2I_d)(\widehat{\Sigma}_1 - \Sigma_1)\|_{\max} + \|(\widehat{\Sigma}_2 + \Sigma_2 - 2I_d)(\widehat{\Sigma}_2 - \Sigma_2)\|_{\max} \right] \\
&\leq \frac{1}{4} \left[ \|\widehat{\Sigma}_1 + \Sigma_1 - 2I_d\|_2 \|\widehat{\Sigma}_1 - \Sigma_1\|_{\max} + \|\widehat{\Sigma}_2 + \Sigma_2 - 2I_d\|_2 \|\widehat{\Sigma}_2 - \Sigma_2\|_{\max} \right]
\end{aligned}
\tag{8}
$$

Follow the classic asymptotic result in univariate OLS and use the union bound, we have

$$
\|\widehat{\Sigma}_1 - \Sigma_1\|_{\max} = O_p(\sqrt{\frac{\log d}{n}}) \quad \text{and} \quad \|\widehat{\Sigma}_2 - \Sigma_2\|_{\max} = O_p(\sqrt{\frac{\log d}{n}}).
\tag{9}
$$

Then, we bound the first operator norm in (8) as

$$
\begin{aligned}
\|\widehat{\Sigma}_1 &+ \Sigma_1 - 2I_d\|_2 \\
&= \|\widehat{\Sigma}_1 - \Sigma_1 + 2\Sigma_1 - 2I_d\|_2 \\
&\leq \|\widehat{\Sigma}_1 - \Sigma_1\|_2 + 2\|\Sigma_1 - I_d\|_2 \\
&\leq d\|\widehat{\Sigma}_1 - \Sigma_1\|_{\max} + 2\|\Sigma_1 - I_d\|_2 \\
&= O_p(\sqrt{\frac{d^2 \log d}{n}}) + O_p(\sqrt{d}),
\end{aligned}
\tag{10}
$$

where the second term of the last equality is due to $\|\Sigma_1\|_2 = O_p(\sqrt{d})$ derived from Assumption 2. Similarly, we have

$$
\|\widehat{\Sigma}_2 + \Sigma_2 - 2I_d\|_2 = O_p(\sqrt{\frac{d^2 \log d}{n}} + \sqrt{d}).
\tag{11}
$$

By plugging (9), (10) and (11) back to (7), we conclude the proof by showing

$$
\|\widehat{\boldsymbol{\xi}}_1 - \boldsymbol{\xi}_1\|_\infty = O_p(r^4 \sqrt{\frac{\log d}{n}} + r^4 \sqrt{d} \frac{\log d}{n}).
$$

$\square$

**A.3 Proof of Theorem 3**

We will work on the space of probability measures on $X \subset \mathbb{R}^d$ with bounded $p$th moment, i.e.

$$\mathcal{P}_p(X) \equiv \left\{ \mu \in \mathcal{P}(X) : \inf_X |x|^p \mathrm{d}\mu(x) < \infty \right\}.$$

The following Lemma follows the Theorem 5.10 in [6], which provides the weak convergence in Wasserstein distance. Hence we omit its proof.

**Lemma 4.** *Let $X \subset \mathbb{R}^d$ be compact, and $\mu_n, \mu \in \mathcal{P}(X)$. Then $\mu_n \to \mu$ if and only if $W_p(\mu_n, \mu) \to 0$.*

Denote $\widehat{W}_p^*(\mathbf{X}, \mathbf{Y}) = \left( \frac{1}{n} \sum_{i=1}^{n} \|\boldsymbol{x}_i - \phi^*(\boldsymbol{x}_i)\|^p \right)^{1/p}$, the empirical Wasserstein distance with true OTM $\phi^*(\cdot)$. The following Lemma follows the Theorem 2.1 in [3] guarantees that $\widehat{W}_p^*(\mathbf{X}, \mathbf{Y})$ is a consistent estimator of $W_2(p_x, p_y)$. We refer to [3] for its proof.

**Lemma 5.** *Under Assumption 2 (a) and (b), $\widehat{W}_p^*(\mathbf{X}, \mathbf{Y})$ converges almost surely to $W_2(p_x, p_y)$ as $n \to \infty$.*

**Proof of Theorem 3.** Notice that, we can decompose the empirical Wasserstein distance as

$$\widehat{W}_p\left( \phi^{(K)}(\mathbf{X}), \mathbf{X} \right)$$
$$= \left\{ \widehat{W}_p\left( \phi^{(K)}(\mathbf{X}), \mathbf{X} \right) - W_p\left( \phi^{(K)}(X), X \right) \right\} + \left\{ W_p\left( \phi^{(K)}(X), X \right) - W_p\left( \phi^*(X), X \right) \right\} + W_p\left( \phi^*(X), X \right)$$
$$\equiv I_1 + I_2 + I_3.$$

First, under Assumption 2 (a) and (b) and with Lemma 5, one can show that $I_1$ converges to 0 almost surely as $n \to \infty$.

For any $k \geq 0$, denote $\Delta^{[k]} = \mathbf{X}^{[k+1]} - \mathbf{X}^{[k]}$. Then, we have

$$\begin{aligned}
\Delta^{[k]} &= (\phi^{(k)}(\mathbf{X}^{[k]}\boldsymbol{\xi}_k) - \mathbf{X}^{[k]}\boldsymbol{\xi}_k)\boldsymbol{\xi}_k^\mathsf{T} \\
&= (\mathbf{Y}\boldsymbol{\xi}_k - \mathbf{X}^{[k]}\boldsymbol{\xi}_k)\boldsymbol{\xi}_k^\mathsf{T} \\
&= (\mathbf{Y} - \mathbf{X}^{[k]})\boldsymbol{\xi}_k\boldsymbol{\xi}_k^\mathsf{T},
\end{aligned} \tag{12}$$

where the second inequality used the fact that $\phi^{(k)}(\cdot)$ is the OTM between $\mathbf{X}^{[k]}\boldsymbol{\xi}_k$ and $\mathbf{Y}\boldsymbol{\xi}_k$.

Therefore, by taking the vector norm to both sides or (12), we have

$$\begin{aligned}
\|\Delta^{[k]}\|_2 &= \|(\mathbf{Y} - \mathbf{X}^{[k]})\boldsymbol{\xi}_k\boldsymbol{\xi}_k^\mathsf{T}\|_2 \\
&= \mathrm{Tr}\{\boldsymbol{\xi}_k^\mathsf{T}(\mathbf{Y} - \mathbf{X}^{[k]})\boldsymbol{\xi}_k\} \\
&= \lambda_k^2 \|\mathbf{Y} - \mathbf{X}^{[k]}\|_2 \\
&= \lambda_k^2 \|(\mathbf{Y} - \mathbf{X}^{[k+1]}) + \Delta^{[k+1]}\|_2 \\
&\geq \lambda_k^2 \left\{ \|\mathbf{Y} - \mathbf{X}^{[k+1]}\|_2 - \|\Delta^{[k+1]}\|_2 \right\} \\
&\geq \lambda_k^2 \left\{ \lambda_{k+1}^{-2} \|\Delta^{[k+1]}\|_2 \right\} = \frac{\lambda_k^2}{\lambda_{k+1}^2} \|\Delta^{[k+1]}\|_2.
\end{aligned}$$

In other words, we have

$$\|\Delta^{[k+1]}\|_2 \leq \frac{\lambda_{k+1}^2}{\lambda_k^2} \|\Delta^{[k]}\|_2 \leq \frac{\lambda_{k+1}^2}{\lambda_0^2} \|\Delta^{[0]}\|_2, \quad \text{for} \quad k \geq 0.$$

According to Theorem 2, $\lambda_k$ is a consistent estimator of the leading eigenvalue of $\Sigma_{\mathrm{SAVE}}$ in the $k$th iteration. Also, according to Theorem 1, $\lambda_k$ is upper bounded by the $k$th eigenvalue of $\Sigma$, almost surely. Then, under Assumption 2 (c), we have $\lambda_k/\lambda_1$ converges to 0 as $d \to \infty$ and $k \geq Cd$ for some $C > 0$. This implies $\|\Delta^{[k+1]}\|_2 \to 0$ as $d \to \infty$ and $k \geq Cd$.

Then, Lemma 4 guarantees that $I_2$ weakly converges to 0 as $d \to \infty$ and $k \geq Cd$ and hence completes our proof. □