[Reviews · NeurIPS 2019]

Reviewer 1



=================================== After Rebuttal =================================== Thank you for the response and the clarifications. We still have some concerns about the complexity of the algorithm and its dependence on the iteration count -- in particular in cases where this may scale with the dimension d based on the specified termination criteria. This being said, it would be great to comment on this in the numerical results and include more timing studies to confirm this dependence. We also strongly suggest that the authors include the extensions listed in the improvement in the final revision. =================================== Before Rebuttal =================================== The authors present a novel algorithm with both theoretical analysis and empirical results. We have a few comments and suggestions for the work: In the introduction, we recommend that the authors also note alternative versions of finding OTM that are not based on solving a linear program (including non-discrete versions of OT based on solving ODEs or finding parametrized maps). In the description of the numerical method, the authors note that the direction that 'explains the highest proportion of variations in the subspace spanned by the current residuals' corresponds to the direction of maximum marginal discrepancy between the distributions. It would be great if the authors could provide more intuition on the correspondence between these two notions and when they may correspond exactly. In the numerical algorithms, we also recommend that they authors indicate the dimensions of introduced variables (e.g., \hat{\Sigma}). Is there a requirement that the sample covariance matrix is not rank-deficient, 2n \geq d, in order to compute the inverse Cholesky factor? Lastly, is there a guideline for how to measure the convergence or select the termination criteria in Algorithm 2. For the computational cost of PPMM, is there an assumption that n >> d that allows for the simplification of the cost of computing the Cholesky factor (otherwise this should add O(d^3) cost)? Could the authors also add a reference or provide more information on the look-up table method that they use to compute the 1D OT map. In the theoretical analysis, we recommend that the authors present assumption 2 as population level conditions on the underlying random variables (e.g., uncorrelated and tail probabilities) instead of conditions on all samples if possible. Could the authors clarify if Assumption 2.(c) must hold for all r or for a specific index. We also believe that indices i and j should be flipped in line 196. Lastly, we'd like to point the authors to recent work entitled 'Minimax rates of estimation for smooth optimal transport maps' by Hutter et. al. for some results on the convergence rates of OTM. For the numerical results, could the authors comment on the non-monotonic convergence of the Wasserstein distance in Figure 3 with increasing iteration count. Do the authors suspect the sliced and random methods to perform differently in all experiments? It would also be great if the authors could provide or report the change in variance of the results over the 100 replications that were used to compute the mean performance. In all experiments the authors also note that they set the number of iterations to scale with the dimension of the OT problem. In practice, did the authors observe continued/stagnating improvement in the error/distances or do the overall maps ever deteriorate (i.e., due to a form of overfitting) with increasing iteration? Lastly, we have a few minor suggestions for the text: correcting the grammar on lines 5,12,17,51,147,267 and 292, clarifying the notation on line 65 (i.e., the map is usually applied to the random variables and not a transformation of the densities), indicate the dependence on Y on line 100 entering through \hat{\phi}, provide a reference that equality is achieved with the Monge map for the Kantorovich problem on line 97, clarifying the phrase 'when the input is two samples' to indicate two sets of samples on line 140, and defining what is meant by 'binary-response sample' on line 143. Lastly, we recommend that they authors clarify how they are applying PPMM for the Google doodle dataset (i.e., expanding lines 307-310 on how the map is constructed for the different categories of images).

Reviewer 2



--- After rebuttal --- --- After rebuttal The authors provided a numerical result for the case with unequal sample sizes. I hope they can provide the details (none in the response, probably due to space limit?) into the revised version as well since this can improve the applicability of this work. For example, how to implement the look-up-table in this case since sorting does not work now. I'm still concerned with the computation of the inverse of Monge map mentioned in Section 4.2 with unequal case. It should also be addressed properly. I think the idea of using project pursuit to find Monge map in high dimensional case is interesting, although it also inherits the issues such as existence and uniqueness from Monge's OT form. I raise my score and incline to accept this paper. --- Original review --- The authors considered a modification of the sliced Wasserstein distance algorithm to compute the Monge map between two sets of points. The novelty is in the application of existing projection-based dimension reduction methods (specifically, SAVE used in this work) to find in each iteration the projection direction that maximizes discrepancy between the (variance of) two projected sets. Numerical experiments show that projection directions selected as such yield faster convergence than the existing ones with random projections etc. Some main comments: 1. The use of selected projection in sliced Wasserstein is interesting, and the computational efficiency looks promising. The reviewer only concerns that the Monge problem is often not well-posed as the Kantoronovich's formulation, and in many cases the Monge map does not exist or is not unique. It may not be that severe in this paper since the problem is basically to find the optimal matching between two equal-sized sets of points, which is a very special problem of optimal transport (OT). The proposed method does not seem applicable to the majority of OT problems considered in machine learning. 2. Could you explain in more details about the assumptions in Line 185-188? In Eq. (1) it appears that the response can be a nonlinear function f of the projected value. But the assumption 1 seems very strong and already excludes that possibility. Your algorithm only allows for the linear case as well. Will this affect your Step 2 in Algorithm 2, especially when the cost in the OT is not defined as (some power) of Euclidean distance? 3. The statements on complexity are a bit problematic: "Suppose that the algorithm converges after K iterations" is not a rigorous assumption for complexity estimate. The number of iteration K should depend on the accuracy of the approximation (e.g., K is a function of epsilon, the accuracy of approximation at the Kth iteration to the true one). From the experiments it does seem that K is proportional to d for the computed W distance to touch the true value, but it's not appropriate to claim K=O(d) when the constant is large and d is small like in your case. 4. It's not explicitly stated, but is \phi in every iteration of Algorithm 2 a permutation? What is the look-up-table in (b) of Algorithm? Is it just sorting? Some suggestions/corrections: Line 29: Better use "\subset" instead of "\in". Algorithm 1 Step 3: should be $I_d$ not $I_p$. Line 196: it should be "... $y_{ik}$ and the $j$th and $k$th component ...". Line 200: it should be "c_l, c_u". Line 202: What is the subscript "p" in the big O? Line 296: The Table 2 with MNIST results is missing from your paper.

Reviewer 3



--------------- Update after reading Author Feedback ------------- I am updating my score to a 6 based on the author feedback. My main concern previously was that the method could only be applied to distributions with equal number of points and uniform weights. In the rebuttal, the authors have shown that the method can be applied to general distributions. My remaining concern is that the algorithm only applies when a Monge map exists which is still a strong restriction that does not cover several applications in registration/partial Procrustes matching. Conditioned on the authors clarifying this in the text, I believe this paper should be accepted. ------------------------------------------------------------------------------ Summary: The authors propose an approximation to the optimal transport problem based on projection pursuit. The algorithm has fast convergence properties compared to similar approaches (for instance, sliced Wasserstein distances) and is easy to implement. Comments: My main concern is that the approach presented in this paper might not be very useful. From what I understand, Algorithm 2 estimates a Monge map between the two distributions. For finite distributions, Monge maps only exist under very restrictive assumptions. It seems that for the algorithm to work, it requires that the initial distributions both be empirical distributions supported on the same number of points. This effectively reduces the problem of computing optimal transport distances between the two distributions to one of computing a minimum cost perfect matching for which efficient (approximate or exact) algorithms already exist. Can this approach be extended to measures p_X and p_Y where X and Y have different cardinalities or where the distributions are not uniform? Beyond this question, I have a few minor comments: The equation between 97 and 98 equates the optimal Monge map to the Kantorovich optimal plan. These are not necessarily equivalent. A Monge map may not exist in many cases. Can you give wall time comparisons for Figure 3? If given the same time budget, how do the two compare? This would be easier to read than Table 1. Is it possible to compare with fast transport solvers, such as the one here: https://github.com/nbonneel/network_simplex In the particular case presented in the paper, comparing to standard perfect matching algorithms would be useful. How does your work relate to the following paper: Paty and Cuturi, "Subspace Robust Wasserstein Distances" In line 166 you mention the computational cost of the method. What is the memory cost? In line 177 you assume that E[X] = E[Y] = 0. I understand that you can assume either X or Y is centered without loss of generality. Why can you assume that both are distributions centered at 0?

[Author Response · NeurIPS 2019]

We would like to thank the meta reviewer and three reviewers for the care with which you handle the submission and for your professional and constructive comments. We have made every effort to address the concerns.

**Response to Reviewer 1's comments**

In the revision, we will review and discuss more OT approaches that are not based on solving a linear program.

Our algorithm utilizes a second-order dimension reduction method to estimate the projection direction. Hence, the leading projection direction corresponds to the direction of the maximum marginal discrepancy between the **variances** of the distributions. Our paper and some dimension reduction literature made bad presentations by ignoring third and higher-order moments. We apologize for this ambiguity and will make it clear in the revision.

(1) In the revision, we will add dimensions to the quantities in algorithms. Yes, we require $2n > d$ for the reason you mention. A viable stopping criterion of Algorithm 2 is to check the angle difference of projection directions between two consecutive iterations. The algorithm is terminated when the angle is close to zero. We appreciate these constructive comments. (2) Yes, the computational cost analysis follows the assumption that $d \ll n^{2/3}$, which allows us to dominate $O(d^3)$ by $O(n^2)$. The lookup table in Algorithm 2 is simply sorting. In the revision, we will explain it clearly and review more literature for the 1d optimal transport and lookup table method. (3) We will revise Assumption 2 with population-level language. We require Assumption 2 (c) to hold for a fixed integer $r > 1$, not every positive integer. We have fixed the typo in Line 196. We acknowledge the enlightening paper you recommend.

(1) The non-monotonic convergence is caused by the non-equal sample means of two point clouds which can cause some troubles to violates the assumptions of SAVE. A remedy is to use a first-order dimension reduction method like SIR to adjust means first. We find it empirically solves the problem. In our experiments, we observe that RANDOM outperforms SLICES in simulations but vice versa for real data. This may suggest that RANDOM is more greedy but less robust. In light of your suggestion, we will report the variance over replications. (2) We do not observe the over-fitting of PPMM in our experiments. The projection direction found by Algorithm 1 tends to converge when $\mathbf{X}^{[k]}$ converges. In contrast, we do observe that RANDOM and SLICED deteriorates in some scenarios.

IMPROVEMENTS: (1) In Algorithm 1, the estimation accuracy of $\widehat{\Sigma}$ depends on the tail probability of $X$ and $Y$. Also, the OTM estimator in Algorithm 2 will be affected by asymmetry and outliers. So we expect the algorithms to perform best when $X$ and $Y$ follow symmetric and sub-exponential distributions (e.g., Gaussian). (2) To converge fast, we require the eigenvalues of $\Sigma$ to decay fast enough (approximately low rank). Hence, the first a few projection directions can explain the majority part of the variance of the discrepancy between two distributions.

**Response to Reviewer 3's comments**

Thank you for this insightful comment. Our method can be extended to the cases of non-equal sample-sizes and non-equal weights with small tweaks. For two point clouds with non-equal sizes, we can use the approximate-lookup table in Algorithm 2. Also, we can calculate a weighted covariance matrix in Algorithm 1 to allow non-equal weights. Here we use a simulated example to demonstrate these cases. We follow a similar setting as in Section 4.1 except that we draw $5,000$ and $1,000$ points from $p_X$ and $p_Y$, respectively. We set $d = 10$ and assign weights to observations randomly. The results are presented in Fig. 1, where the colored lines are the sample means of estimated Wasserstein distances over 100 replications and the black dashed line is then calculated by the "short simplex" method, which serves an oracle. In addition, the average wall time (until converge) is: PPMM(0.3s), RANDOM (1.4s), SLICED10 (14s) and "short simplex" (74s). In the revision, we will discuss this important extension with additional numerical justifications. We believe that such an extension will make the proposed algorithm applicable to a much larger family of OT problems. We also plan to discuss the extension of the algorithm to Kantoronovich's formulation.

(1) The conditions in Assumption 1 are widely used in dimension reduction literature and are not as restrictive as they seem. It will not prohibit a nonlinear model. Intuitively, (a) and (b) assumes that $u^T Z$ behaves like Normal. (2) We agree with your comment on complexity statements and will re-write this part accordingly. Yes, $\phi$ is a permutation, and the lookup table step is just sorting. We will make corrections according to your minor comments. In Line 202, $O_p$ stands for order in probability which is similar to $O$ but for random variables. (3) In the revision, we will make a comprehensive discussion of assumptions and how they affect the convergence analysis.

**Response to Reviewer 4's comments**

Our algorithm can be extended to address your concern about non-equal sample-sizes and non-equal weights. Please see our response 6 to Review #3 for more details and a simulation result. We appreciate this important comment.

(1) In the revision, we will report the wall time. (2) Also, we will compare our method with fast transport solver and standard perfect matching. (3) The paper from Paty and Cuturi is very interesting and helpful. We will cite and discuss it in the revision. (4) The memory cost is $O(nd + d^2)$. We will discuss it in the revision. We are grateful for these constructive comments.

The assumption in line 177 is for mathematical simplicity. It can be removed if we use a first-order dimension reduction method like SIR to adjust means before we apply SAVE. Our theory can be readily applied to this case. We thank the reviewer for pointing out this insightful issue.

Figure 1: Simulation with the point clouds with different sample-size and non-equal weights

[Meta-Review · NeurIPS 2019]

Reviewers have read the rebuttal with interest and have slightly raised their scores. The paper provides a nice addition to the current literature dealing with the estimation of W using projections onto subspaces, such as https://arxiv.org/abs/1902.00434 https://arxiv.org/abs/1901.08949 https://arxiv.org/abs/1903.03784 which the paper needs to refer and discuss if it were published. The paper is on the fence. minor comments: - in the intro you describe accurately that GANs involve computing a transport map (l.21), and provide several examples. Then, in l.27, this becomes computing an optimal transport map. These two things are however different. - l.33 "These methods, however, are not able to provide the explicit 34 form of the OTM". I am not sure the method proposed here by the authors does so as well, so there is a logical problem here. In any case, any solver outputting a coupling can generally output a suboptimal map through the so-called "barycentric projection", this has been used several times. - "The composition of all the one-dimensional maps serves as the final estimate of the target OTM" I am not sure I understand what is meant by composition. Isn't it rather a sum? - "In addition, the existing projection-based approaches usually suffer from slow convergence or even not convergent". Not sure what is meant here by convergence. - I am not sure I understand the point of section 4.1, since Sliced W was never presented as something defined to approximate OT in dimensions higher than 2 (and neither is the approach presented here in fact). Wouldn't it be better to present it as a form of implicit regularizer? Numbers in table 1 do not strike me as meaningful.